# The Microbiome as Part of the Contemporary View of Tuberculosis Disease

**DOI:** 10.3390/pathogens11050584

**Published:** 2022-05-16

**Authors:** Martín Barbosa-Amezcua, David Galeana-Cadena, Néstor Alvarado-Peña, Eugenia Silva-Herzog

**Affiliations:** 1Laboratorio de Farmacogenómica, Instituto Nacional de Medicina Genomica (INMEGEN), Mexico City 14610, Mexico; mvz.mba@gmail.com; 2Laboratorio de Inmunobiología y Genética, Instituto Nacional de Enfermedades Respiratorias (INER), Mexico City 14080, Mexico; david.microbiomics@gmail.com; 3Coordinación de Infectología y Microbiología, Instituto Nacional de Enfermedades Respiratorias (INER), Mexico City 14080, Mexico; nestor321.alvarado@gmail.com; 4Laboratorio de Vinculación Científica, Facultad de Medicina-Universidad Nacional Autonoma de Mexico-Instituto Nacional de Medicina Genomica (UNAM-INMEGEN), Mexico City 14610, Mexico

**Keywords:** tuberculosis, microbiome, dysbiosis, disease dynamics, disease severity

## Abstract

The study of the microbiome has changed our overall perspective on health and disease. Although studies of the lung microbiome have lagged behind those on the gastrointestinal microbiome, there is now evidence that the lung microbiome is a rich, dynamic ecosystem. Tuberculosis is one of the oldest human diseases, it is primarily a respiratory infectious disease caused by strains from the *Mycobacterium tuberculosis* Complex. Even today, during the COVID-19 pandemic, it remains one of the principal causes of morbidity and mortality worldwide. Tuberculosis disease manifests itself as a dynamic spectrum that ranges from asymptomatic latent infection to life-threatening active disease. The review aims to provide an overview of the microbiome in the tuberculosis setting, both in patients’ and animal models. We discuss the relevance of the microbiome and its dysbiosis, and how, probably through its interaction with the immune system, it is a significant factor in tuberculosis’s susceptibility, establishment, and severity.

## 1. Introduction 

Tuberculosis (TB) is a disease that has accompanied humankind for thousands of years [1,2]. Signs of the disease have been found in Egyptian mummies from 2400 BC, and detailed descriptions of it exist in Chinese and Greek literature, including Hippocrates in 400 BC and Galen in 200 AD [3].

TB, caused by the organisms of the *Mycobacterium tuberculosis* Complex (MTBC), includes *Mycobacterium tuberculosis*, *M. africanum*, *M. orygis*, *M. bovis*, *M. microti*, *M. canetti*, *M. caprae*, *M. pinnipedi*, *M. suricattae*, and *M. mungi*, has been responsible for over one billion deaths in the last 200 years [4]. Pre-COVID-19 pandemic, TB was globally the deadliest infectious disease, claiming 1.4 million lives in 2019 and causing illness in close to 10 million. It ranks even now among the top thirteen causes of death worldwide [5]. Furthermore, the health care disruptions caused by the pandemic in 2020 led, for the first time in years, to an increase in deaths by TB with millions of undiagnosed and untreated cases [5].

Infection with *M. tuberculosis* (MTB) occurs when the aerosol droplets carrying the bacillus are inhaled. However, not everyone infected becomes sick. Only a small proportion (5–10%) of immunocompetent individuals will develop active TB (ATB); many will clear the pathogen, and others will resolve in a latent tuberculosis infection (LTBI). LTBI individuals have no symptoms, are unable to transmit the disease, but can revert to active TB at any point in their lives. The heterogeneous manifestation of MTB infection suggests a decisive role of the host in the progression of the disease. The host’s innate and adaptive immunological mechanisms, and their interaction with the microbiome, influence the balance between pathogenesis and host clearance [6,7,8,9,10].

The advent of Next Generation Sequencing (NGS) has revealed the significant role of the microbiome in the balance between health and disease; it has been proposed that changes in the microbiome may become a powerful biomarker for many pathological conditions in the near future [11,12,13,14]. Although the study of the microbiome of the respiratory tract has lagged behind that of other body sites, mainly due to the invasiveness and difficulty in obtaining reliable samples, it has become clear that: (1) the lower respiratory tract (LRT) is not sterile; (2) acute and chronic respiratory diseases change the ecological conditions of the respiratory tract, thus affecting the resulting microbial communities; (3) the microbiome trains the immune system; and (4) the immune system modulates the microbiome [15,16,17,18,19]. This interaction of the immune response and the microbiome is critical to the balance between health and disease, including the response to pathogens and other challenges such as allergies, asthma, cystic fibrosis, and cancer [18,20,21,22,23]. This review aims to provide a brief overview of the lung microbiome and its relation to TB with its clinical manifestations.

## 2. The Clinical Course of Tuberculosis

Clinically, TB presents as a disease with a subacute to chronic evolution caused by infection with MTBC. Although MTBC can infect many parts of the body, the vast majority of infections (84%) reside in the lungs as pulmonary TB [5]. The primary infection takes place mainly in the alveoli, where alveolar macrophages phagocytose MTB. It then either crosses the alveolar barrier by diapedesis to settle in the interstice, or spreads directly by migration through the alveolar barrier into circulation, leading to a systemic spread [24]. This primary infection can have at least three outcomes: First, clearance of MTB by the immune system, either by innate or acquired immune response without memory of T cells; although some individuals will clear the pathogen and preserve a robust memory T cell response. In a second outcome, MTB is not cleared but persists in a latent state (LTBI), defined as the state of continuous immune response to MTB antigens but with no evidence of clinical manifestations or bacterial replication [4,25]. The third outcome involves the progression to active disease (ATB) or subclinical TB, characterized by bacterial growth, rapid host deterioration, and leads to different degrees of clinical manifestations [26]. 

MTB bacilli are cloistered in a granuloma, the histopathology stamp of TB; it is composed of macrophages, lymphocytes, and other immune cells in response to lingering stimuli. The granuloma is very important for containing the infection; there is a constant clash of pro-inflammatory and anti-inflammatory signals. The result of this either promotes or limits the spread of MTB. If there is a strong pro-inflammatory response in this process, then a remodeling of the granuloma with liquefaction and softening of the caseum, as well as the destruction of the lung parenchyma, may signal the beginning of ATB [27]. On the other hand, a predominantly anti-inflammatory response within the granuloma is associated with a decreased risk of reactivation and better clinical outcomes [28,29].

The World Health Organization (WHO) estimates that about a quarter of the world’s population is infected with MTB; however, only 127 new cases per 100,000 population were reported in 2020, which suggests that there are millions of people with LTBI functioning as a reservoir for the disease [5]. If left untreated, approximately 5–10% of these LTB infections will progress to active TB during their lifetime. Therefore, the diagnosis and treatment of LTBI are paramount for controlling and eliminating TB. Individuals with LTBI can progress to active TB disease, or it remains as latent tuberculosis infection, depending on the changes in host immunity, host microbiome, and other risk factors that include HIV infection (Relative Risk (RR) 18), undernourishment (RR 3.2), alcohol abuse disorders (RR 3.3), diabetes (RR 1.6), and smoking (RR 1.6) [4,5].

### 2.1. Definitions and Clinical Manifestations

Clinically, weight loss and night sweats have the most relevant association with active TB, with an odds ratio of 4.47 and 3.29, respectively [30]. However, common symptoms include cough, fever, anorexia, and chest pain [31], all common to many respiratory illnesses, and thus cannot be used for TB diagnostics. This is why TB diagnosis must be confirmed by culture and molecular diagnostic tests [4]. Although a persistent cough is not a definite diagnosis, it is one of the most common symptoms of advanced pulmonary ATB. As the disease progresses, increased inflammation is followed by tissue necrosis that can progress into the tubercular caverns, which are regions with a high bacillary load. The inflammation of the lung parenchyma close to the pleura can cause pleuritic pain [32]. Dyspnoea can be a significant clinical component after a substantial amount of the lung is destroyed or there is a significant pleural effusion [30]. Physical examination of the chest in pulmonary TB is unrevealing [33]. However, the changes are more pronounced in the upper lobes because MTBC is strictly aerobic, and these areas are more ventilated, leading to greater growth of the bacilli [34].

Extrapulmonary Tuberculosis (EPTB) refers to any bacteriologically confirmed case of TB involving organs other than the lungs, e.g., pleura, lymph nodes, abdomen, genitourinary tract, skin, joints, bones, or meninges [35]. It represents 16% of all tuberculosis cases. Its development depends on age, presence, or absence of underlying disease, the MTB strain, immune status, and ethnic background, and, possibly, the microbiome [5,36]. About 10–50% of EPTB patients have associated pulmonary TB [37].

Without treatment, TB is a life-threatening disease. Studies in patients with pulmonary TB, and positive smear microscopy, prior to the advent of anti-TB drugs, were followed up for five years: 50–60% died; 20–25% were cured spontaneously; and 10–25% continued with symptoms of TB [38].

### 2.2. Tuberculosis Treatment

The objective of any TB therapy is, first, to reduce the number of actively growing bacilli in the patient, thereby decreasing the severity of the disease, and halting transmission of MTB; second, to eradicate populations of persisting bacilli to achieve a long-lasting cure and prevent relapse, and third to prevent the acquisition of drug resistance during therapy [39].

The treatment of ATB relies on multidrug regimens. In the case of drug-susceptible TB (DS-TB), the treatment includes six months of four first-line anti-TB drugs: isoniazid (H), rifampicin (R), ethambutol (E), and pyrazinamide (Z) [40]. This treatment is divided into two phases: an intensive or bactericidal phase with the four drugs H, R, E, Z, administered for two months, with the objective of reducing the bacillary load and the transmission, as well as avoiding the selection of resistant strains associated with these four drugs. The second, or sterilization, phase includes R and H administered for four months, this phase aims to continue with the sterilization of the tissue, including intracellular bacilli, prevent relapses, and therefore have a cure [39]. This regimen has proven to be very successful, with an 85% success rate, and has been widely adopted worldwide for decades [5]. Currently, it is possible to shorten the treatment from six to four months with a scheme with similar efficacy and safety, that includes Rifapentine (P), Moxifloxacin (Mfx), H, and Z [41].

Antibiotic resistance is a great concern for all infectious diseases, including TB. Drug-resistant TB (DR-TB) has increased from 30,000 cases in 2009 to 157,903 in 2020 worldwide [5,42]. There are several types of DR-TB: Rifampicin-Resistant (RR), bacteria resistant to Rifampicin; Multidrug-Resistant TB (MDR-TB), those resistant to at least isoniazid and rifampicin; Pre-Extreme Drug-Resistant (Pre-XDR) are MDR, as well as to any fluoroquinolone; and XDR-TB are strains that fulfill the definition of Pre-XDR for at least one drug of the WHO’s Group A list [43] (see below).

The treatment of DR-TB (MDR, Pre-XDR, XDR) can be either with standardized regimens recommended by WHO or individualized plans that are tailored to the pattern of resistance and the patient’s particular characteristics, in which specific drugs can be modified according to the pattern of resistance [44]. Anti-tubercular drugs have been classified based on efficacy into Group A: Levofloxacin (Lfx), Moxifloxacin (Mfx), Bedaquiline (Bdq) and Linezolid (Lzd); Group B: Clofazimine (CFZ), Cycloserine (Cs) or Terizidone (Trd); and Group C: Ethambutol (E), Delamanid (Dlm), Pyrazinamide (Z), Imipenem-cilastatin (Imp-Cln) or Meropenem (Mpm), Amikacin (Am) or Streptomycin (S), Ethionamide (Eto) or Prothionamide (Pto) or P-aminosalicylic acid (PAS) [45]. On the other hand, treatment of LTBI has several options: these include six to nine months of daily H or one month of daily Rifapentine plus Isoniazid and four months of daily Rifampicin, just to mention the more common options [25,46].

In sum, all TB treatment involves long multidrug regimens that undoubtedly will have profound effects on the microbiome and the host’s overall wellbeing.

## 3. Human Microbiome and Its Importance in Health and Disease

The term host-associated microbiome refers to the microbial communities occupying a discrete habitat as well as their ’theater of activity’, which result in the formation of individual ecological niches. The microbiome forms a dynamic ecosystem that is integrated with its eukaryotic host [47,48]. To fully understand this interaction in the balance between health and disease, a systems approach, that includes proteomic, metabolomic, and genomic data of the distinct microbiomes, will be necessary.

The factors that have been proposed to contribute to the formation of this host-associated microbiome include evolutionary conserved relationships between the host and the colonizing microorganisms [49], interactions between members of the microbial community [50], and with the immune system [51]. Furthermore, structure and distinct physicochemical properties may develop ecological niches with recognizable functional profiles [52]. When all these factors are balanced, or in homeostasis in a particular niche, the microbiome is said to be in eubiosis [53,54], which is a state that reflects a microbiome resilient to changes and thus benefits both the host and the microbial communities (Figure 1a). Given the confusion that the terms balance and unbalance can cause, in the present review we propose to define dysbiosis as ‘the reduction of adaptive capacity of a eubiotic microbiome to changes in physicochemical conditions, immune response, microbial diversity, keystone taxa (taxa that are highly connected with other microorganisms and can significantly influence the structure and function of the microbiome) [55,56], dominance or function, increase in pathobionts (commensal microorganisms that have the potential to cause disease), that cause unfavorable alterations for the host or contributes to disease’ (Figure 1b). It should be clarified that an infection is only one example of dysbiosis, since any significant change in the microbiome that affects its function is a dysbiotic state, including those due to metabolic alterations [57].

### 3.1. Microbiome Functions

After millions of years of coevolution, the microbiome is able to perform the critical functions of many biological processes of the host, including modulating the metabolic phenotype, regulating epithelial development, and modulation of the immune response. In metabolism, it facilitates the digestion of complex macromolecules [58] and vitamin synthesis [59]. The commensal microbiome has been proposed to prevent the establishment of new microorganisms by competitive exclusion [60], changing the physicochemical factors of the microenvironment [61,62], producing antibiotics and secondary metabolites [63], or modulating the expression of virulence factors [64]. Indirectly, through its metabolites, the microbiome may stimulate the development and function of the immune system [19,65]. Furthermore, both in the gastrointestinal and respiratory tract, the microbiome induces secretion of mucins by Goblet cells that protect the epithelia, and Paneth cells in the gut produce antimicrobial molecules [21]. The microbiome may also promote resistance to future infections in the gastrointestinal tract through the production of metabolites that promote inflammation, which in turn contributes to the protection against future pathogen invasions, which has been called meta-organism memory [66].

### 3.2. Gut–Lung Axis

Human bodies are made up of different systems that are in constant communication to maintain homeostasis, despite physical barriers. Similarly, microbiomes of different niches have long-distance effects on other body organs, including the skin, gut, brain, and lung [67,68,69,70]. This review will focus on the relationship between the gastrointestinal (GI) and respiratory systems. The gut has the most concentration of microorganisms in the human body; it is for this reason, and the fact that the samples are more easily accessible, that the gut is the most studied site regarding the host microbiome. Gut microorganisms come from food and water intake [71,72] and are seeded at birth [73]. In healthy individuals, gut microbiota are dominated by Firmicutes (e.g., *Lactobacillus*, *Bacillus*, and *Clostridium*), as well as Bacteroidetes (e.g., *Bacteroides*), and to a lesser extent, Proteobacteria (e.g., *Escherichia*), and Actinobacteria (e.g., *Bifidobacterium*) [74]. As mentioned above, the effect of the gut microbiome is not limited to the GI tract. It can extend to other organs, including the lung, in what is known as the “gut–lung axis” [8]. Similarly, the lung microbiome impacts the gut microbiome, and presumably establishes a truly bidirectional network of communication [74]. This communication is achieved through the microorganisms’ metabolic products, including small chain fatty acids (SCFAs), which modulate the immune response in both gut and lung systems [70]. Quorum sensing, which allows intraspecies, interspecies, and interkingdom cell-to-cell communication, has been associated with colonization, regulation of virulence factors, resistance to antibiotics, and the adaptive capacity to changes in the microenvironment for the communities that comprise the microbiome [75].

### 3.3. The Lower Respiratory Tract (LRT) Microbiome

The LRT microbiome changes over time, as well as between individuals [76]. In healthy lungs, microbial communities are primarily determined by immigration, elimination, and reproductive rates, whereas in advanced lung disease, membership is primarily determined by regional growth conditions and reproduction rates [15]. Nevertheless, there is individual compositional microbiome stability and possibly an individual core LRT-commensal microbiome [77]. Similar to the gut microbiota, healthy lungs are predominantly comprised of the phylum Bacteroidetes, Firmicutes, and Proteobacteria, followed by lesser proportions of Actinobacteria [16], but at the genus level, the most abundant are *Streptococcus*, *Prevotella*, *Fusobacterium*, *Haemophilus*, *Pseudomonas*, *Veillonella*, and *Porphyromonas* [78].

The lung microbiome has been reported to change in different conditions including metabolic diseases [79], asthma [80], COPD [81], pulmonary cystic fibrosis [82], and cancer [83]. During infections, these changes can be produced by the entry of a pathogen, an increase of pathobionts, loss of commensals or keystone taxa [55,56]. In the case of the entry of a pathogen, the microbiome may, together with the host’s immune response, eliminate the pathogen and maintain the eubiosis [15], or go into a state of dysbiosis which can result in disease [84]. The microbiome can protect from secondary infections inducing IgA and IgG specific responses and adaptive immune response [85]. Nevertheless, dysbiotic microbiomes can also favor co-infections, as in the case of Respiratory Syncytial Virus (RSV), where the modified microenvironment allows the expansion of pathobionts [86]. Thus, dysbiosis of the respiratory microbiome is a critical element in systemic inflammatory responses and the clinical outcome of patients [87].

Recasting the system’s approach, where we consider that all microbiomes are interconnected, LRT infections affect the gastrointestinal tract. Influenza, a primary respiratory infection, may cause digestive tract manifestations through hematogenous dissemination of infected lymphocytes from the respiratory tract [88]; and a decreased Bacteroides/Firmicutes ratio in the GI tract has been observed during RSV respiratory tract infection [89]. On the other hand, gut dysbiosis has been associated with both decreased levels of butyrate and exacerbated bacterial pneumonia, which supports the critical role of SCFAs in pulmonary host defense [90] and increases susceptibility to infections [91,92].

## 4. Microbiome Changes during Tuberculosis

Although dysbiosis has been reported to have negative health effects [93], and was associated with the pathogenesis of various diseases: gastrointestinal diseases, obesity, diabetes, allergies, asthma, colorectal cancer, etc. [13,94], its influence on MTB infection in the lungs is still a subject of study [95].

### 4.1. Microbiome and Mycobacterium Tuberculosis Infection

As discussed above, MTB infection can have a spectrum of clinical manifestations, ranging from clearance of the bacillus to active establishment of the infection. What determines these outcomes is poorly understood, but has been primarily associated with host factors, such as the immune system response [96] and, more recently, the microbiome [9,97,98].

Although some authors have reported differences in the microbiota between healthy individuals and patients with active TB [99,100,101,102], the primary pulmonary response to MTB colonization is very difficult to assess directly on humans, which is why the use of animal models has been employed. These models have provided valuable information, increasing our knowledge of the disease.

Studies on aerosolized MTB-infected mice, showed a rapid loss (6 days) of intestinal microbial diversity followed by a gradual recovery of beta-diversity, probably because of crosstalk between the microbiome and the host immune system during TB infection [103]. However, similar studies observed slower (12 weeks) and less evident alterations in the intestinal microbiota of mice after the infection with MTB, probably due to differences in the MTB strain used (CDC1551 vs. H37Rv) and/or genetic factors between the animal models (Balb/c vs. C57BL/6 mice) [104].

Parallel studies using murine models of gastrointestinal dysbiosis induced by broad-spectrum antibiotics prior to MTB inoculation, show increased bacilli colonization and dissemination (liver and spleen). This dysbiosis was associated with a reduction in the number of mucosal-associated invariant T cells (MAIT), less expression of IL-17A, IFN-γ, and TNF-α (associated with protection against TB) and increased regulatory T cells (associated with susceptibility to TB); additionally more and larger pulmonary granulomas were observed in these mice, suggesting that antibiotic-induced dysbiosis increases the spread of the disease [9,97]. Furthermore, the restoration of the microbiome through fecal transplant reversed these effects: it increased the number of MAIT cells, the expression of IFN-γ and TNF-α (produced by MAIT cells Th1), and reduced the regulatory T cells, supporting a key role for the microbiome in the colonization of the lungs, the response to MTB, and the severity of the infection in mice [9,97].

Taken together, these findings demonstrate that microbial communities are essential for the modulation of host immunity and that changes in the microbiome, even at distal sites, can determine TB outcomes and prognosis. However, the precise role of dysbiosis in the balance between health and disease is just beginning to be understood.

### 4.2. The Microbiome during Latent and Active TB

As mentioned earlier, the immune system controls the infection of approximately 90% of people exposed to MTB; these individuals either completely clear the bacilli or remain asymptomatic throughout their lives as LTBI [80]. In LTBI, the immune response restrains MTB within granulomas, where the bacteria may persist, but not spread. It is possible that the lung microbiome is involved in the formation and dynamics of the granuloma, probably through the stimulation of the Th1 response through IL-17, and it is a dysbiotic state that influences the progression of the disease [105]. The influence of the microbiome on the host’s adaptive immune response has been reported in other respiratory infections such as influenza, where an intact gut, and/or nasal microbiome is necessary to induce Th1 cytotoxic T lymphocytes (CTL) and IgA responses during viral infection [19].

The role of the GI or LRT microbiome in TB progression is not yet fully understood. However, Perry et al. [106] reported that patients with LTBI and *H. pylori* infection, (one of the most prevalent pathogenic gastric bacteria in the world) had a better Th-1 cytokine response (INF-γ, IL2, TNF-α, CXCL-10) to TB antigens, compared to LTBI individuals with no *H. pylori* infection. In addition, non-human primates exposed to TB as well as individuals with LTBI are less likely to develop active TB when they have a prior *H. pylori* co-infection. This suggests that *H. pylori* infection generates a pro-inflammatory state that enhances the host’s innate immune response against MTB and other infectious diseases. Conversely, MTB inoculation after natural colonization of the intestine of mice by *H. hepaticus*, in combination with an intestinal dysbiosis characterized by a greater abundance of *Bacteroidaceae* and reduction of *Clostridiales*, *Ruminococcaceae*, *Lachnospiraceae,* and *Prevotellaceae*, cause an overstimulation of the innate immune response and excessive inflammation (increased pro-inflammatory cytokines) that increased the susceptibility to MTB, and severe lung damage [107].

Other studies, working with a non-human primate model and a combination of 16S rRNA and metagenomics, found an enrichment of the families *Lachnospiraceae* and *Clostridiaceae*, even before infection, in the gut microbiome of monkeys that developed severe TB. The prevalence of these bacteria continued after MTB infection with an added reduction of *Streptococcaceae*, *Bacteroidales* RF16, and *Clostridiales vadin* B660 [14]. Furthermore, studies in West Africa where both *M. africanum* (MAF) and MTB are endemic, showed that patients with TB due to MAF had lower alpha diversity, increased *Enterobacteriaceae* in the GI tract, and higher expression of inflammatory genes prior to antibiotic treatment, when compared to the MTB patients and healthy controls. In addition, the MAF patients had a reduction in *Actinobacteria* and *Verrucomicrobia* when compared to the MTB patients. The authors speculate that in this region, where an individual can encounter both bacilli, which bacteria (MTB or MAF) will establish an infection is determined by the host’s immune system and its microbiome [108]. This further supports the hypothesis that the gastrointestinal microbiome modulates the susceptibility and development of TB.

On the other hand, studies on the LRT microbiome of TB patients have shown variable results when compared to healthy individuals, perhaps due to differences in samples (BAL vs. sputum), populations analyzed, experimental design, and the definition of healthy. However, several authors have reported an increased microbial diversity in the lower respiratory tract of ATB patients [100,101,102,109,110]. Other studies have shown increased diversity in DR-TB vs. DS-TB patients [102,111].

This increased microbial diversity during ATB may be due to tissue damage reduction of lung commensal bacterial and a higher susceptibility to opportunistic microorganisms such as members of the *Leptotrichia*, *Granulicatella*, *Campylobacter*, *Delfitia* or *Kingella genus*; or pathogens such as *Klebsiella*, *Pseudomonas* and *Acinetobacter*, which have been associated with other respiratory tract pathologies [109,111], and may contribute to additional damage and aggravated symptoms. In fact, epidemiological studies have shown a correlation between opportunistic infections and increased risk of DR-TB development [112], probably due to an indiscriminate use of antibiotics.

Thus, in addition to multiple risk factors (diabetes, malnutrition, co-infections, parasites, etc.) [113], there is clear evidence that supports the crosstalk between the microbiome and the immune system in the establishment of MTB infection, and between microbiome dysbiosis and progression of MTB infection.

### 4.3. Microbiome Changes during and after Antituberculosis Treatment

As aforementioned, the standard treatment for drug-susceptible TB requires the use of broad-spectrum and specific antibiotics (H, R, Z, and E) against mycobacteria for at least six months, causing intestinal dysbiosis that persists in patients for more than a year after finishing the treatment [104]. In fact, rifampicin, a broad-spectrum bactericide, causes the greatest alterations in the microbiome [114].

As mentioned previously, there is an increase in the incidence of antibiotic-resistant TB (DR-TB) [5], whose treatment can be up to 20 months and involves the use of combinations of antibiotics that induce intestinal dysbiosis during and for up to eight years after treatment [93]. Fecal transplantation and the use of probiotics have been proposed for the restoration of microbiome eubiosis after DR-TB treatment to reduce the development of comorbidities and poor outcomes [93].

Oral administration of *Lactobacillus rhamnosus* NK210 and *Bifidobacterium longum* NK219 partially help to restore the populations of Firmicutes, Bacteroidetes, Proteobacteria, and Verrucomicrobia in a murine model of gut dysbiosis caused by the use of ampicillin, and during a state of LPS-induced systemic inflammation. In both cases, the administration of NK210 and NK219 decreased the expression of IFN-γ, TNF-α, Tbet; it increased the expression of IL-10 and Foxp3 (both involved in the reduction of the inflammatory response), improving gut dysbiosis and decreasing inflammation [115]. However, the inoculation of a single microorganism was not enough to restore the normal microbial community or prevent recurrent infections in patients with other diseases, such as intractable bacterial vaginosis, but a microbiome transplant from healthy donors was effective in improving symptoms and the laboratory features of the disease [116].

### 4.4. Influence of the Microbiome in Post-TB Patients

Lung damage, reduced pulmonary function, and proinflammatory lung microenvironment in post-TB patients make them more susceptible to develop recurrent respiratory infections by bacteria (*P. aeruginosa*, *H. influenzae*, *M. catarrhalis*, and *S. aureus*) and fungi (*A. fumigatus*, *A. niger* and *A. flavus*) [111,117].

Furthermore, approximately 6% of patients who complete the standard treatment for drug-susceptible TB, relapse [118]. The persistent dysbiosis of the lung microbiome of TB patients has been associated with treatment failure and relapse [80,119]. Relapsing patients show differences in alpha diversity with an increase in the phyla Proteobacteria and Actinobacteria (rich in pathogenic species) and a reduction in Bacteroidetes (mainly beneficial commensal organisms) in the gut microbiome [80]. Notably, a higher *Pseudomonas*/*Mycobacterium* and lower *Treponema*/*Mycobacterium* ratio in the lung microbiome may be a risk factor associated with relapse [119].

These data suggest that maintaining microbiome eubiosis could be beneficial for TB recovery, as well as to avoid relapse [80]. However, more studies are needed to establish the connection between the microbiome and poor TB outcome [120].

## 5. Conclusions and Perspectives 

The study of the microbiome has changed the perspective of the interactions between microorganisms and their host, as well as our understanding of health and disease. As we have stressed in this review, the microbiome has a central role in the normal physiology of the host, as well as in the immune response before and during infections. An important point to consider is that this interaction is dynamic. The elements that surround and form any particular microbiome are constantly changing and it is the adaptive capacity of an eubiotic microbiome that maintains the balance and wellbeing of the host.

Studies of the microbiome in respiratory disease are recent but have shown that the microbiome has an important role in the establishment and progression of the disease. In particular, TB and microbiome studies are only starting to understand this relationship. TB is an ancient disease that is still now, in the XXI century with new diagnostics and treatments, having a devastating impact on millions of people. The COVID-19 pandemic exposed the fragility of our healthcare systems and left TB patients without diagnosis and treatment. It made clear that new strategies for diagnosis and treatment are desperately needed; we think the microbiome study may provide new insights.

Although further studies are required to fully understand the interaction between the microbiome, the immune response, and MTB pathogenesis, preliminary studies show a possible association between dysbiosis, susceptibility to MTB infection, and TB progression. Dysbiosis generated by changes in the lung environment of TB patients, including loss of commensal and keystone taxa, allows the colonization and proliferation of oral, upper respiratory tract, and environmental microorganisms, resulting in opportunistic infections that aggravate the disease and maybe a risk for relapse (Figure 2).

Furthermore, increased severity of the disease was shown in animal models that were previously treated with antibiotics, and the susceptibility of individuals to different members of the MBTC was associated with distinct gut microbiome.

As in any other infectious disease, antibiotics induce a rapid loss of bacterial populations, generating a dysbiosis that persists even after treatment ends. Restoration of the microbiome at the end of antibiotic treatment could benefit the patient. In this sense, the use of probiotics capable of modulating the immune response and reducing inflammation could help restore eubiosis, avoiding reinfections and relapses.

It is tempting to think that the microbiome, with its interaction with the immune response, determines the clinical spectrum of the disease, as was suggested for other respiratory infections. The role in immune modulation of fungi, viruses, and parasites in the pathogenesis of TB must also be analyzed. Future, longitudinal studies on the interaction of the respiratory and gastrointestinal microbiome of tuberculosis patients and their close contacts can identify biomarkers to better understand the establishment and progression of tuberculosis and improve patient prognosis.

## Figures and Tables

**Figure 1 pathogens-11-00584-f001:**
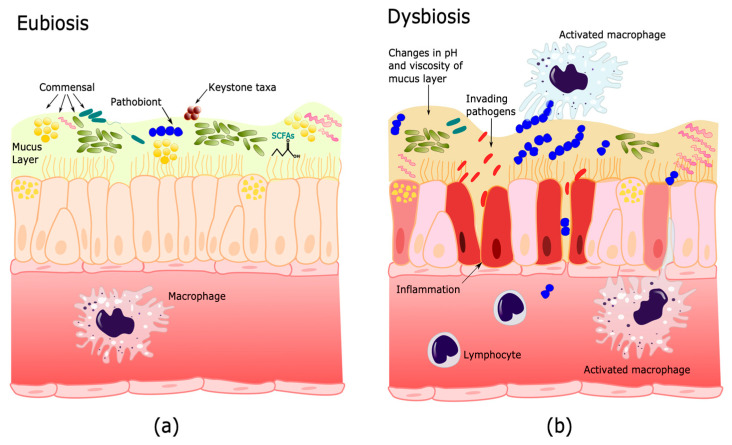
Microbiome dynamics. In eubiosis (**a**), the factors that conform to the microbiome are in homeostasis and produce metabolites that favor the host’s wellbeing. While in dysbiosis, (**b**), microorganisms can decrease their adaptive capacity to changes produced by an invading pathogenic agent and microenvironments that promote an increase in pathobionts, changes in the inflammatory response, and the immune system. Elaborated with Inkscape.

**Figure 2 pathogens-11-00584-f002:**
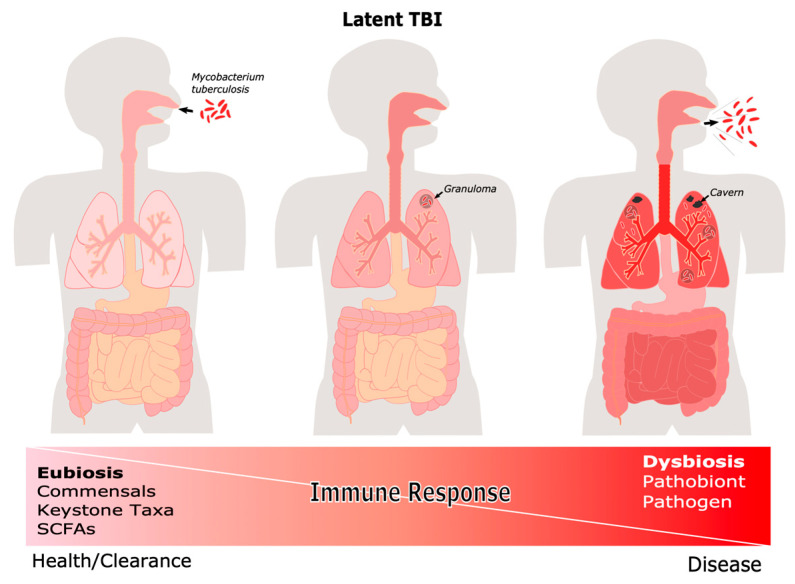
**Host-associated microbiome factors in the TB spectrum of disease.** TB presents itself as a spectrum of disease, after infection the individual may clear the bacilli, become LTBI, or develop ATB. The outcome of TB infection is modulated by the microbiome as well as the host. (SCFAs, Short Chain Fatty Acids).

## Data Availability

Not applicable.

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
