# Peer review of "The Microbiome as Part of the Contemporary View of Tuberculosis Disease"

_pathogens, 2022, doi:10.3390/pathogens11050584_

Round 1

Reviewer 1 Report

Very timely and worthwhile review, very well done. THANKS!

Author Response

Thank you very much for your comments

Reviewer 2 Report

This review by Barbosa-Amezcua et al. is well written, and the concepts are highly pertinent to the tuberculosis field. It gives a complete picture of the disease and highlights the relevance of the microbiota’s interplay in the outcome. Find below some suggestions to improve the work.

1.-Line 29. Please, define the MTBC and all the species it comprises.

2.-Line 184. The terms keystone taxa and pathobiont are introduced in the legend of figure 1 and the figure itself, but the definition is not provided until in lines 237-240. It would be helpful to have the definition earlier.

3.-It is customary that figures appear after being mentioned in the text. Figure 2 appears before its reference in the text.

4.-The coinfection TB+COVID19 is barely covered in this review. Considering the impact of the microbiota in COVID19, its potential impact on the coinfection is worth discussing. See example references below.

Gut microbiota composition reflects disease severity and dysfunctional immune responses in patients with COVID-19.

Yeoh YK, Zuo T, Lui GC, Zhang F, Liu Q, Li AY, Chung AC, Cheung CP, Tso EY, Fung KS, Chan V, Ling L, Joynt G, Hui DS, Chow KM, Ng SSS, Li TC, Ng RW, Yip TC, Wong GL, Chan FK, Wong CK, Chan PK, Ng SC.

Gut. 2021 Apr;70(4):698-706. doi: 10.1136/gutjnl-2020-323020. Epub 2021 Jan 11.

PMID: 33431578

The human microbiome and COVID-19: A systematic review.

Yamamoto S, Saito M, Tamura A, Prawisuda D, Mizutani T, Yotsuyanagi H.

PLoS One. 2021 Jun 23;16(6):e0253293. doi: 10.1371/journal.pone.0253293. eCollection 2021.

PMID: 34161373

The hygiene hypothesis, the COVID pandemic, and consequences for the human microbiome.

Finlay BB, Amato KR, Azad M, Blaser MJ, Bosch TCG, Chu H, Dominguez-Bello MG, Ehrlich SD, Elinav E, Geva-Zatorsky N, Gros P, Guillemin K, Keck F, Korem T, McFall-Ngai MJ, Melby MK, Nichter M, Pettersson S, Poinar H, Rees T, Tropini C, Zhao L, Giles-Vernick T.

Proc Natl Acad Sci U S A. 2021 Feb 9;118(6):e2010217118. doi: 10.1073/pnas.2010217118.

PMID: 33472859

Author Response

Reviewer 2:

This review by Barbosa-Amezcua et al. is well written, and the concepts are highly pertinent to the tuberculosis field. It gives a complete picture of the disease and highlights the relevance of the microbiota’s interplay in the outcome. Find below some suggestions to improve the work.

We greatly appreciate your valuable comments and suggestions, which have strengthened the manuscript.

1.-Line 29. Please, define the MTBC and all the species it comprises.

We added the species that comprise the Mycobacterium Tuberculosis Complex in lines 29-30

2.-Line 184. The terms keystone taxa and pathobiont are introduced in the legend of figure 1 and the figure itself, but the definition is not provided until in lines 237-240. It would be helpful to have the definition earlier.

We moved the definition of keystone taxa and pathobiont to the first time they are mentioned in lines 176 -179, which improves the understanding of the concepts.

3.-It is customary that figures appear after being mentioned in the text. Figure 2 appears before its reference in the text. 

Figure 2 and its legend were moved after the paragraph where it is mentioned to line 427.

4.-The coinfection TB+COVID19 is barely covered in this review. Considering the impact of the microbiota in COVID19, its potential impact on the coinfection is worth discussing. See example references below.

This is a very important and timely comment. We agree with the reviewer that respiratory infections, particularly that of SARS-CoV2, have an important impact on the respiratory microbiome and are critical in understanding susceptibility and development of tuberculosis, as we discuss briefly in the ‘Conclusions and Perspectives’ section. However, at this point, we thought it was out of the scope of this review to dive into COVID-19 and Tuberculosis co-infection and focused only on tuberculosis and respiratory microbiome. In fact, some of us are preparing a manuscript on the impact of COVID-19 on the upper respiratory tract microbiome in Mexican patients.

Reviewer 3 Report

General comment:

This review is well organised and covers a state of the art topics in comprehensive way. It is easy to understand the current situation about respiratory microbiome and tuberculosis. It will be very useful for the readers to inspire the updated concept of lower respiratory systems and infections.

Specific comments:

  1. Line 28: AC must be a typo of AD.
  2. Line 139: RR-TB means the tuberculosis patient with bacteriologically proven rifampicin resistant Mycobacterium tuberculosis, defining the resistance detected by WHO endorsed rapid test (mainly Xpert MTB/RIF). It does not mean a bacterium resistant only to rifampicin. The rifampicin resistance is only proven and others are unknown.

Author Response

Reviewer 3

General comment:

This review is well organized and covers a state of the art topics in a comprehensive way. It is easy to understand the current situation about respiratory microbiome and tuberculosis. It will be very useful for the readers to inspire the updated concept of lower respiratory systems and infections.

We thank the reviewer for the comments, which have improved the manuscript.

Specific comments:

  1. Line 28: AC must be a typo of AD.

Thank you, we missed it.

  1. Line 139: RR-TB means the tuberculosis patient with bacteriologically proven rifampicin resistant Mycobacterium tuberculosis, defining the resistance detected by WHO endorsed rapid test (mainly Xpert MTB/RIF). It does not mean a bacterium resistant only to rifampicin. The rifampicin resistance is only proven and others are unknown.

Thank you, we removed the word ONLY in line 141 to acknowledge this, and keep with WHO definition.